# Technology-Enhanced Airport Services—Attractiveness from the Travelers' Perspective

**Márk Miskolczi [1,*], Melinda Jászberényi [1] and Dávid Tóth [2]**

1   Institute of Marketing, Corvinus University of Budapest, H-1093 Budapest, Hungary; jaszberenyi@uni-corvinus.hu

2   Faculty of Business and Economics, University of Pécs, H-7622 Pécs, Hungary; david.toth@munich-airport.de

*   Correspondence: mark.miskolczi@uni-corvinus.hu

**Abstract:** The rapid emergence of automation brings new opportunities for airport development. Airports strive to maximize passenger satisfaction as well as optimize their operation. However, the lack of knowledge of consumer preferences might be an important barrier to achieve these objectives. Therefore, our study aims to unveil the potential of service development alternatives based on artificial intelligence (AI). For this, a systematic literature review (SLR) and a quantitative analysis of a survey have been conducted. The results of the empirical research are based on 593 responses; most of the subjects belong to generation Z (digital natives) and Y (millennials). The analysis revealed attitudes towards different AI-based transport solutions and AI robots that provide information at the airports. Based on the perceived attractiveness of such services, the environmentally conscious behaviour of consumers, and sociodemographic data, subjects were classified into three different clusters (Negligents, AV Enthusiasts, and Robot Fanatics). Results proved the attractiveness of AI-based transport services that can be used in the air-side zone. Among the millennials, the idea of self-driving buses running between airport terminals is the most appealing. Greater interest in AI-based communication solutions can be perceived among generation Z. For both generations, environmentally conscious consumption is also of paramount importance. The attractiveness of AI-based solutions has been analyzed in a tourist environment, which might be a good starting point for further research into the technology acceptance of AI-based services.

**Keywords:** airport service development; self-driving airport buses; AI robots; cluster analysis



## 1. Introduction

Aviation provides greater access to destinations, thus expanding the dimensions of the tourism sector. International airports nowadays are no longer just facilities for taking off and landing aircraft. There are more than 40.000 airports worldwide [1] offering complex travel experiences and serving billions of passengers every year. The largest hubs have been undergoing constant spatial and functional alterations, thus developing into commercial centers [2]. Industry 4.0 has also great potential in improving travel processes and therefore the tourist experience [3]. Technological advances (e.g., digitalized services, automation, etc.) provide many novel business opportunities, but also create dilemmas related to airport development.

Previous studies [4,5] consider the digitalization and automation of services as amplifiers of competitive but sustainable airport operations. Researchers [6,7] emphasize that airport attractiveness is greatly enhanced by the development of self-service technologies (SSTs) (e.g., automatic immigration control, self-boarding, online booking changes, etc.) due to the decreased need for HR assistance, paper-based solutions, and the increased individuality of passengers.

Industry 4.0. has created artificial intelligence (AI)-based services, which have radically transformed communication (e.g., application of chatbots) [8] as well as passenger transport opportunities (e.g., the emergence of autonomous vehicles) [9] in other areas of life.

Based on these phenomena, our research question concerns how consumers perceive the use of AI-based services at airports. The three main objectives of our research are to reveal the attitude of passengers towards AI-based airport services (1), to identify the alternatives in which AI-based services could be successfully applied based on consumers' feedback (2), and also to sort the respondents into different groups to provide a comprehensive picture of the segment (3). For these objectives, both secondary (systematic literature review—SLR) and empirical research (questionnaire among tourists with aviation experience) were conducted.

The survey pointed to a positive attitude towards the adaptability of self-driving buses between terminals, especially among generation Y (millennials). Results also shed light on the potential use of other services based on self-driving vehicles in the air-side and land-side zones of the airport (e.g., experience driving, self-driving shuttle services), as well as on the judgement of AI-based information robots. Based on consumers' attitudes towards the technology, environmental awareness, and socio-demographic characteristics, three clusters (Negligents—dominated by generation Z, AV Enthusiasts—dominated by generation Y, and Robot Fanatics—dominated by generation Z) have been identified. Results showed that as the age progresses, the attractiveness of self-driving technology increases, while the intention to use AI robots decreases.

The paper is structured as follows: since airports have shifted from conventional stations to complex service centers, Section 2 introduces both land-side and air-side services of airports and the basic definitions of technology innovations, i.e., highlighting the background of automation (AI-based) technology and its applicability with some examples at the airport environment. To prove transparency, Section 3 draws up our hypotheses and research methods (SLR, cluster analysis) applied during the research. Research outcomes are presented in Section 4, which reveals the adoption of AI-based services among respondents. Our conclusions, further research directions (Sections 5 and 6), and the limitations of our research (Section 7) are presented at the end of the study.

## 2. Airport Services and Digital Solutions—Systematic Literature Review

### 2.1. Key Definitions of Airport Services

Airports can be divided into two areas in terms of operation [7]: the land-side zone, which is available without limitations (e.g., parking lots, check-in area, etc.), and the air-side zone, which is only for passengers and staff after the security screening [9]. It is important to make a distinction, since different services and, therefore, different developments might be needed in both zones.

It is a trend nowadays that larger airports' air-side zones are turning into a shopping centre crammed with shopping facilities, restaurants, and other kinds of amusements [9]. This is a response to the increasing mass of tourists flying [10]. In addition to traditional airport services, an increasing number of special experience elements have already appeared, which also greatly influence the image of the destination [11]. Changi Airport in the Far Eastern metropolis of Singapore is voted the best in the world due to its special experience offer (e.g., four-storey high slide at Terminal 3, Cactus Garden, Forest Valley, etc.) [12]. Munich Airport (Germany - Munich) is the only one in the world to have a brewery, which also offers traditional Bavarian food [13]. Archaeological finds discovered during the construction of the terminal can be visited at London Heathrow Airport (UK - London) [14]. Due to technological advances, automated and digital solutions for airport services have appeared in both zones, which enables the performance of tasks without human assistance [15,16]. In aviation terms, these services are called self-service airport solutions (SSTs) (e.g., online check-in, baggage check-in, ticketing kiosks, etc.) that both simplify airport processes and adapt to individual consumer preferences [17]. At the same time, SSTs can reduce waiting times significantly [18].

IATA forecasts that the role of automated processes will reach 50% of all administration over the next few years [19]. Studies [17,19] emphasize that SSTs elevate the travel experience to a higher level, and thus have a positive impact on the general perception of

the airport operation as well. However, studies examined changes in consumer satisfaction in the context of a narrower range of SSTs, primarily the impacts of check-in processes and baggage drop-offs [20,21]. The adoption of automated airport services based on artificial intelligence is mainly mentioned as an important further research direction [22]. Since transport and communication solutions operated by artificial intelligence correspond to the general definition of SST services [18], in our research, we classify them as STT services, although they appear less in this context in previous research. Section 2.2 clarifies the basic concepts related to technology.

### 2.2. Emerging Technologies

The fourth industrial revolution (Industry 4.0) has been transforming the entire value chain of all sectors due to the appearance and spread of machines suitable for partial or complete replacement of human resources [23]. Industry 4.0-created artificial intelligence (AI) results in an increasing integration between digital devices and humans [24]. AI is the part of robotics that can perform tasks that previously required human assistance [25]. AI-based robots can learn and understand complex situations. The main difference between "traditional" and AI-based technology is that the former was intended to perform only repetitive tasks, but AI-based programs have limited ability to replace more complex human functions [26]. Two of the most promising innovations that perhaps best show the different applications of the technology are AI robots and self-driving (i.e., autonomous, driverless) vehicles [27].

By self-driving vehicles, different means of transport can be distinguished whose control is partially or completely removed from human control; the role of the driver can be taken over by artificial intelligence [28]. Under the SAE framework [29], currently available vehicles are at level 2 of automation (partial automation). The technology has emerged in many areas of passenger transport (e.g., individual vehicle ownership—Tesla vehicles [30]; shared mobility—Waymo services [31]; public transport—minibuses by Navya Arma [32]).

As automation solutions are an emerging field of transport innovation, they might affect airport services in the near future. The literature on airports [9,27] mentions self-driving vehicles as a sustainable transport option, as these vehicles are electric and can lead to the reduction and/or redeployment of human resources [27,28]. Self-driving vehicles can be used to transport passengers between terminals (air-side application) as well as to approach the airport (land-side application).

Research into driverless autobuses confirms a positive attitude towards the technology due to its perceived environmental (e.g., energy-saving, minimization of damage to the flora) and psychological (e.g., avoidance of human error) benefits [29–31]. Research [33–35] suggests that AI-based technologies may be attractive primarily to the young segment (especially to gen Z) and can be used as a virtual tour guide (e.g., AutoTour), and so they can replace the conventional ways of sightseeing (Hop-on-Hop-off services). Self-driving minibuses have been tested at Charleroi Airport near Brussels in collaboration with Flibco shuttle [36] service provider in 2019. The vehicle called Navya is electric (operates for 9 h continuously) and can transport 15 people (11 seated, 4 standing) at a maximum speed of 25 km/h [36]. In 2020, due to the continuous expansion of passenger traffic, Luxembourg Airport [4] also has introduced a transit service between the two terminals. In the pre-epidemic period, the airport predicted a significant increase in the number of passengers in the future, so increasing the traffic between the two terminals. According to previous plans, self-driving airport buses would be adopted between Brussels and Brussels Zaventem Airport in 2021 at the earliest, after tests are completed successfully. However, the outcomes of the development are uncertain due to the COVID-19 pandemic.

Another important stage in the development of AI-based solutions is the application of robots in communication. AI-based chatbots are already used to complement administrative processes (e.g., airline information interfaces), and the technology has also begun to spread in airport communications. In 2018, an AI-robot was tested at the Munich Airport's Terminal 2 and provided passengers with information on timetables and other

travel information (e.g., restaurants and shops) [37]. The robot called "Josie Pepper" is capable of learning and provides individual answers to each question. Josie is currently under development [37]. Other similar examples for AI robots: Troika at Incheon Airport in Seoul [38], Spencer, the robot guide at Amsterdam-Schiphol Airport [39], or Robird at Canadian Airports [40].

Although our research examines only the consumer perception of AI-based mobility and communication alternatives, other applications should also be mentioned. The use of AI-robots is becoming more common in other airport operations (e.g., logistics—transport of goods, restaurant tasks, luggage carrying, etc.) [41,42]. An outstanding example is Singapore Airport, which employs around 21,000 people [43]. However, as local labour supply is decreasing, and the number of tourists is constantly increasing, it will no longer be possible to manage the airport operations without AI-robots [43,44].

Despite the increasing use of AI-based airport services, the social aspects are still under-researched according to our literature review.

- Articles on SST service development are extensive, but they focus on a narrow segment of digital services and tend to disregard the potential of AI-based solutions.
- Even though more and more airports are testing the technology, the applicability and technology-acceptance of AI-based services (e.g., self-driving autobuses) has been less studied. The review confirmed the positive attitude of potential consumers, but previous studies did not analyse the phenomena in the context of airport development.
- The attractiveness of artificial intelligence-based robots with anthropomorphic (human-like) traits has also been under-researched in the airport-related literature. The limitations of technology diffusion have been addressed in several studies, but, from the user's point of view, applicability comes to the fore less often.
- We can see that airports have been working to improve the airport stay experience (e.g., Munich Airport) as well as to simplify processes with automated services (e.g., Charleroi Airport). Based on the trends, it is conceivable that services based on artificial intelligence may appear in both the air-side and land-side zones, as well as provide entertainment and practical functions.

Consequently, as these are still open questions, we identify a research gap and a need to explore the applicability of different AI-based services based on consumers' attitudes.

## 3. Research Design and Methods

### 3.1. Data Collection

Systematic literature review (SLR) is an exploratory research process to synthesize and critically estimate enquiries into a specific topic according to a pre-defined perspective [45]. Existing research has been collected to identify research gaps, which demand more attention and determine our empirical research plan. Our literature review aims to interpret the basic characteristics of emerging technologies and their relevance in airport service development. As Figure 1 shows, the SLR process comprises three phases, which begin with a systematic search. The first step was criteria selection, in which keywords ($n = 4$), scope of the research (peer-reviewed, published in the last ten years), and database (Google Scholar) were selected. According to titles, research yielded a total of 46 relevant sources. In the second phase, the number of relevant articles was decreased by the exclusion criteria. Finally, 31 relevant sources were identified and analysed qualitatively. Information published on the official website of each airport was also used for the analysis ($n = 9$).

Table 1 shows the distribution of the analysed sources by topic, year of publication, data used, and keywords. The elaboration of each topic was balanced by the number of sources, and we relied more on the latest (published in the last two years) literature, which ensures the actuality of the results. Much of the literature is based on quantitative data collection ($n = 14$), mainly related to SST and airport satisfaction. Results from the literature review are also outstanding ($n = 9$).

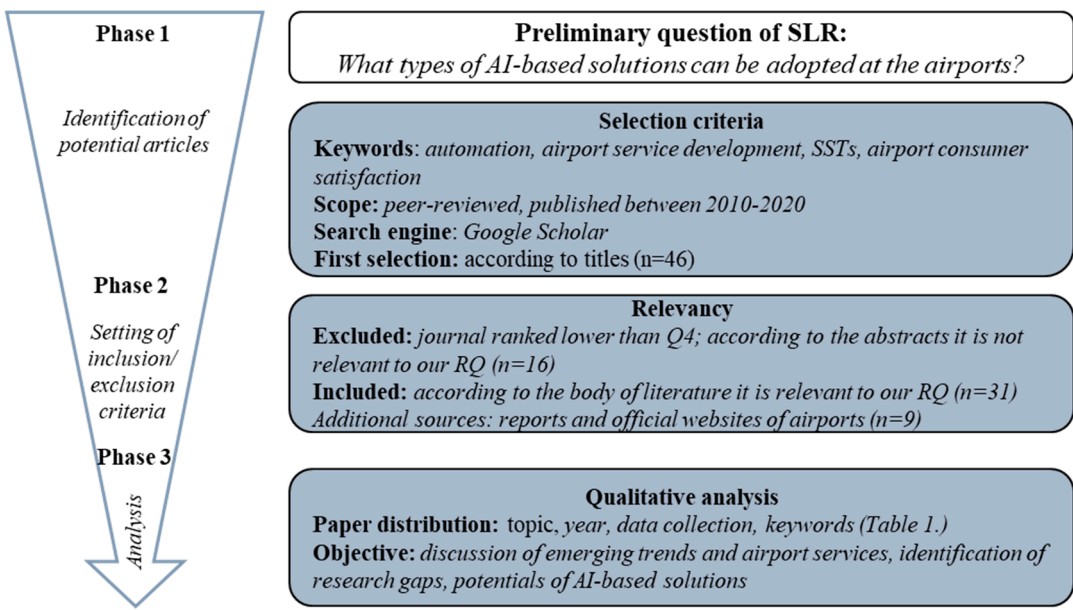

**Figure 1.** Research process. Source: authors' own editing.

**Table 1.** Description of the literature analysed. Source: authors' own editing.

| Characteristics | Number of Secondary Sources |
| --- | --- |
| Distribution by topic | |
| Airport operation development | 7 |
| Airport and tourism | 5 |
| SSTs at the airports | 7 |
| Automation—self driving/driverless/autonomous vehicles at the airports | 6 |
| Airport travellers' satisfaction | 6 |
| Distribution by year of publication | |
| 2018–2020 | 16 |
| 2014–2017 | 8 |
| 2010–2013 | 7 |
| Distribution by data | |
| monograph | 2 |
| literature review | 9 |
| stakeholder interviews | 1 |
| online survey | 14 |
| online content analysis | 1 |
| data analysis (statistical treatment) | 3 |
| focus group interviews | 1 |
| Distribution by keywords applied | |
| airport SSTs | 6 |
| digitalization (artificial intelligence) | 5 |
| automation airport | 6 |
| airport consumer satisfaction | 6 |
| airport service development | 8 |

### 3.2. Data Collection

As most articles analysed by SLR examine consumer attitudes in the form of an online questionnaire (Table 1), this data collection method has also been chosen in our empirical research. Online surveys are cost-effective, easy to conduct, and contain a wide range of question styles [46]. Passenger satisfaction and airport service expectations have commonly been captured through online surveys. The findings from such studies [20,34]

can identify prospects for service improvements and encounter some unique aspects of travellers' attitudes towards novel technologies.

As previous studies have not fully analysed consumer attitudes towards AI-based solutions, there is a lack of understanding of how travellers evaluate AI-based services at the airport. Therefore, our survey aims to explore general consumer habits (e.g., preferred means of transport and activities during an airport stay, the importance of environmental protection) (1) and attitudes towards AI-based technologies (2), especially on self-driving transport and AI robots (3).

### 3.3. Analysis Method

According to the research directions, six hypotheses have been conducted (Table 2).

**Table 2.** Hypotheses. Source: authors' own editing.

| No. | Hypothesis | Explanation |
|---|---|---|
| H1 | AI-based land-side transport-related services are attractive among tourists. | Owing to the development of self-driving technology, (partly) autonomous vehicles have been becoming more widely available. Using the following hypothesis, we examine whether subjects would like to approach the airport with vehicles equipped with such functions. |
| H2 | AI-based air-side transport-related services are attractive among tourists. | Based on the literature review, self-driving buses between terminals are already tested. This solution is complemented by evaluating other AI-based transportation alternatives to enhance the social aspects of the applicability of the technology. |
| H3 | AI-robots are more attractive than AI-based transport services among tourists. | By answering the hypothesis, we answer which AI-based services (communication or transport-related) are worth developing. |
| H4 | There is a correlation between "Gender" and the attractiveness of AI-based services. | Through an exploratory market segmentation conducted by cluster analysis, we explore the correlation between the basic socio-demographic variables, the environment-consciousness, and the perceived attractiveness of AV-based solutions. |
| H5 | There is a correlation between "Age" and the attractiveness of AI-based services. | |
| H6 | There is a correlation between "Sustainable attitude" and the perceived attractiveness of AI-based services. | |

For answering these questions, statistical analysis methods (e.g., descriptive statistics, correlation analysis—Pearson's Chi-Square Test) were adopted. The basic part of the analysis was a cluster analysis, which was used to segment research subjects.

With cluster analysis, the observation units can be arranged into relatively homogeneous groups based on the variables involved in the analysis [47]. The process is considered successful if the units are similar to their group peers but different from the elements of the other groups [48]. Cluster analysis is often used for market segmentation or to examine sales opportunities for a new product or service [49]. The limitation of cluster analysis is that no conclusions can be drawn from the sample for the population, and so it can be used primarily as an exploratory technique [48]. In the case of the following research, this is not an exclusive factor, as the sampling is primarily based on generation Z and Y, and our objective is to show the attitudes towards AI-based solutions among these age groups.

### 4. Results

### 4.1. Sampling

Data for this study were collected between September and October 2020 via Qualtrics Online Survey Software. The target population was adults 18 years or older. Subjects without any aviation experience were excluded from the survey. Travelers must have flown at least once before the COVID-19 period. Respondents included in the research were asked to rate their flight experience and attitudes toward each technology regardless of the

COVID-19 pandemic. Ethical requirements were met (e.g., anonymity, possible refusal to answer certain questions).

### 4.2. General Characteristics of Respondents

A total of 623 answers were received. 30 answers were excluded from the analysis due to incorrect completion. Overall, 593 usable responses were analyzed. The number of sample items exceeds the expected size of 200 of exploratory marketing research [48], and so the conclusions can be valid and applied for further research. Based on gender, the majority of respondents are female (53.79% of the total sample), but the sample is relatively balanced.

The age distribution of the respondents was determined according to the generational marketing theory [50] which distinguishes five categories (Figure 2.). Based on age groups, the dominance of Generation Z (digital natives) and Generation Y (millennials) can be seen (71.70% of the total sample) (Figure 3).

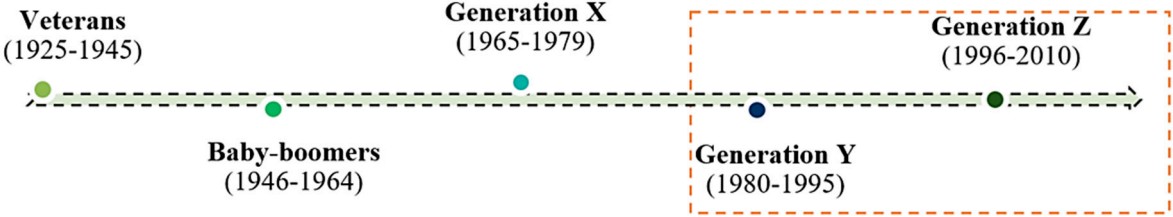

**Figure 2.** Theory of generational marketing. Source: Authors' own editing based on [50].

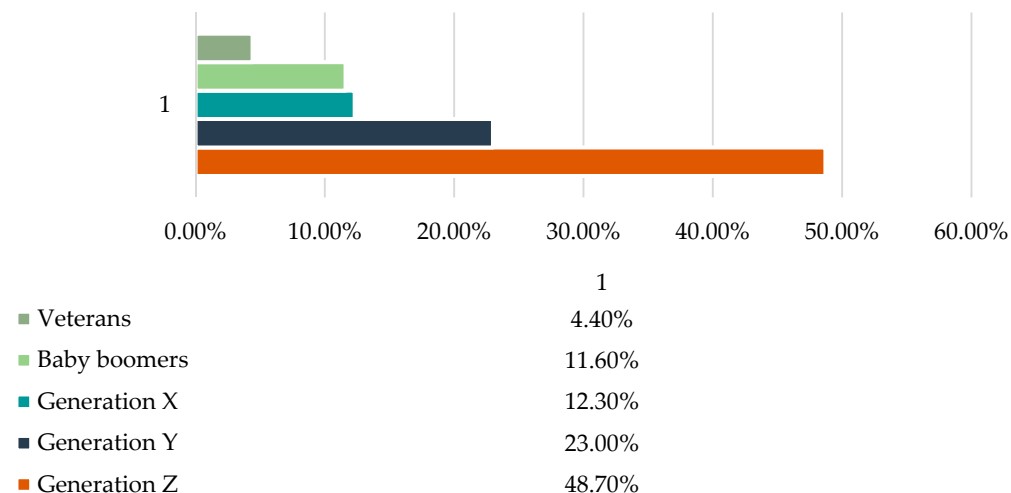

|   | 1 |
|---|---|
| ■ Veterans | 4.40% |
| ■ Baby boomers | 11.60% |
| ■ Generation X | 12.30% |
| ■ Generation Y | 23.00% |
| ■ Generation Z | 48.70% |

**Figure 3.** Proportion of respondents based on generations. Source: Authors' own editing.

As far as the other sociodemographic variables are concerned, the vast majority of respondents have a high school diploma (46.31%), and 40.7% of respondents have university degrees. Due to the sensitivity of the issue, the financial status of the respondents was not determined based on a salary range. Respondents were asked to classify themselves on the basis of self-assessment into the social class with which they can best identify. Consequently, the middle class is overrepresented (61.67%) in the sample (see details: Table A1 in the Appendix A).

### 4.3. General Characteristics of Consumer Habits

During the analysis, we focused on two airport-related consumer traits. On the one hand, the preferred modes of transport used to get and leave the airport have been revealed (Table 3).

**Table 3.** Ranking of means of transport (%). Source: authors' own editing.

| Means of Transport | % | Ranking | Type of Activities | % | Ranking |
|---|---|---|---|---|---|
| Public transport | 31.62% | 1. | Relaxing, entertainment | 35.27% | 1. |
| Shuttle bus | 28.1% | 2. | Catering | 30.36% | 2. |
| Taxi | 15.52% | 3. | Shopping | 24.58% | 3. |
| | | | Work | 7.52% | 4. |
| Privately-owned car | 12.95% | 4. | Others: reading a book, studying, window-shopping, | 2.27% | 5. |
| Rent a car | 8.55% | 5. | | | |
| Shared mobility (e.g., car-sharing; carpooling) | 1.52% | 6. | | | |

Based on the results, a ranking was created among the modes of transport preferred by subjects. The largest group of respondents (31.62%) approach the airport by public transport, followed by shuttle service (22.51%), taxi (18.52%), and privately-owned cars (15.95%). The proportion of rental car use (8.55%) has also appeared, which further strengthens the importance of road transport alternatives. Respondents had to evaluate which are the main activities that determine their airport stay (Table 3). Respondents spend most of their time during their stay at the airport with passive activities (e.g., relaxation, entertainment—listening to music, etc.) (35.27%). The importance of restaurants (33.36%) and shopping (21.58%) is also authoritative. The role of other activities is negligible (below 10%).

Consequently, respondents prefer the pre-organized and public (e.g., public transport, shuttle service) mobility solutions to other means of transport. As far as spending time at the airport is concerned, entertainment and recreation opportunities play a primary role.

*4.4. Attitudes towards AI-Based Technologies*

Respondents were asked to assess the perceived attractiveness of AI-based solutions and the importance of environmental protection on a seven-point scale (1 = "not attractive/important at all"; 7 = "very attractive/important") (Table 4).

**Table 4.** AI-based service evaluations. Source: Authors' own analysis with IBM SPSS Statistics 25.

| | | N | Mean | Std. Deviation | Skewness | | Kurtosis | |
|---|---|---|---|---|---|---|---|---|
| | Code | Stat. | Stat. | Stat. | Stat. | Std. Error | Stat. | Std. Error |
| AI-based solutions in general | - | 593 | 5.5943 | 1.68167 | −0.415 | 0.100 | −0.802 | 0.199 |
| Self-driving shuttle bus between the city and the airport | LS_1 | 593 | 4.4508 | 1.80593 | −0.225 | 0.100 | −0.852 | 0.199 |
| AutoTour with self-driving vehicles | LS_2 | 593 | 4.2415 | 1.95333 | −0.147 | 0.101 | −1.153 | 0.201 |
| | LS_1_2 | | Σ 4.3461 | | | | | |
| Self-driving bus between the terminals | AS_1 | 593 | 5.1067 | 1.78342 | −0.696 | 0.100 | −0.486 | 0.199 |
| Self-driving cars as amusement (experience driving) | AS_2 | 593 | 4.2768 | 2.15021 | −0.135 | 0.102 | −1.391 | 0.203 |
| AI-based robots for communication | AS_3 | 593 | 3.9422 | 1.90938 | 0.130 | 0.101 | −1.087 | 0.201 |
| | AS_1_2_3 | | Σ 4.4419 | | | | | |
| Importance of sustainability | ST | 593 | 5.0101 | 1.45993 | −0.539 | 0.099 | −0.469 | 0.198 |
| Valid *N* (listwise) | | 593 | | | | | | |

In the analysis, we tested the idea of different types of service based on artificial intelligence among the respondents:

- Self-driving shuttle bus between the city and the airport: As the use of shuttle services is prominent among respondents (28.51%), we examined how they would relate to the introduction of a self-driving shuttle service.
- AutoTour with self-driving vehicles: During the literature review, we came across the idea of a tourism service specializing in self-driving vehicles (AutoTour—AI as a tour guide), the attractiveness of which we considered worth examining in an airport context.
- Self-driving bus between terminals: This service is being developed and tested, which makes it necessary to explore social attitudes.
- Self-driving cars as amusement (experience driving): Based on the wide range of unique experiences that some airports offer, and the fact that a significant number of respondents seek entertainment (35.27%) during their stay at the airport, the opportunity to try out self-driving cars can also be a potential service element.
- AI robots for communication: the literature confirmed that technology is being developed and tested, so it is necessary to explore social attitudes.

Each AI-based service was grouped according to whether it is related to airport air-side (AS_) or land-side services (LS_). Mean values represent the central tendencies of variables. The overall attractiveness of automation technology is relatively positive (Mean: 5.5943). If the standard deviation (Std. Dev.) is greater than 2, observations within the variables would be so scattered that it would also affect the stability of the estimation [48]. In all cases of examined variables, Std. Dev. is below 2. In the case of specified AI-based services, the greatest attractiveness is shown by the self-driving buses that can be used between terminals (Mean: 5.1067). The least attractive service is the AI-based robots for information gathering (Mean: 3.9422).

Kurtosis and skewness provide information on the distribution of variables. Distribution of land-side variables examined is considered non-normal, since the variables "LS_1 and LS_2" indicate a slight left-skewed distribution (−0.225; −0.147) with a flat kurtosis (−0.852; −1.153). Distribution of air-side variables is also considered non-normal, since the variables "AS_1" and "AS_2" indicate a negatively skewed distribution (−0.696; −0.135) with a flat kurtosis, while the variable "AS_3" indicates a positive (right-skewed) (0.130) distribution. Respondents were asked how important environmental protection (e.g., selective waste management, avoidance of plastic consumption, etc.) is to them. According to the figures, variable "ST" is not normally distributed either (negatively skewed distribution, negative kurtosis) but proves the relatively strong importance of sustainability among respondents (Mean: 5.0101).

Among the AI-based transport solutions examined, the attractiveness of buses running between terminals stands out. The attractiveness of the AI-based robot lags behind in transport solutions minimally. Consequently, in the next phase of the analysis, we examined which consumer segments can be identified based on these two variables AS_1—Attractiveness of self-driving buses between the terminals and AS_3—Attractiveness of AI-robots for information gathering at the airport.

### 4.5. Cluster Analysis

In the next phase, hierarchical and non-hierarchical cluster analysis methods were used based on each other. The hierarchical method can be advantageous when the number of sampling units is high, and the results obtained are less dependent on the outliers [49].

Before the analysis, we examined the correlation between the two variables included in the analysis (AS_1; AS_3). If the correlation coefficient of the two variables is very high (above 0.9), it is advisable to exclude one from the analysis, as this can lead to distortions and redundancies [47]. Based on Pearson's Correlation, the correlation is significant at the 0.01 level (2-tailed); the relationship between the variables is in the negative medium strength (−0.329) area, which is below 0.9. Therefore, cluster analysis

can be performed. The analysis is based on variables measured on the same scale (1–7), and so no standardization of variables is required.

First, we determined the ideal number of classes and the centres, using a hierarchical technique. Subsequently, the observation units were grouped by a non-hierarchical method based on the cluster centres derived from the hierarchical method. The Ward method is a common method of variance in which the mean of all variables is calculated for each cluster, and then the squared Euclidean distance is calculated for each observation unit. For each step, we combined the two clusters with the smallest increase in standard deviation within the cluster. During the cluster analysis, the distance between the elements was examined.

During conducting the Ward method, the groups that increase the standard deviation within the cluster the least are aggregated (Table 5). There is a big jump when merging the last two clusters (the coefficient increasing from 1368.282 to 2527.559). The increase in coefficients during the previous merger can also be considered significant (coefficients: stage 596: 875,907 and stage 597.: 1,368,559). Based on the calculation, two or three clusters can be formed.

**Table 5.** Cluster analysis—Agglomeration Schedule. Source: Authors' own analysis with IBM SPSS Statistics 25—EXCERPT.

| Stage | Cluster Combined | | Coefficients | Stage Cluster First Appears | | Next Stage |
|---|---|---|---|---|---|---|
| | Cluster 1 | Cluster 2 | | Cluster 1 | Cluster 2 | |
| 591 | 7 | 10 | 325,601 | 589 | 577 | 597 |
| 592 | 11 | 25 | 380,663 | 584 | 582 | 595 |
| 593 | 8 | 16 | 436,343 | 565 | 586 | 594 |
| 594 | 6 | 8 | 512,412 | 587 | 593 | 598 |
| 595 | 11 | 12 | 606,251 | 592 | 583 | 596 |
| 596 | 4 | 11 | 875,907 | 590 | 595 | 597 |
| 597 | 4 | 7 | 1,368,282 | 596 | 591 | 598 |
| 598 | 4 | 6 | 2,527,559 | 597 | 594 | 0 |

The increase in coefficients is also represented in Figure 4. The horizontal axis shows the number of merging steps (Stage), while the vertical axis shows the Values (coefficients). Larger fractures can be observed on the graph, which can be considered as an elbow criterion. Based on the elbow criterion, we examined the two- and three-cluster solutions. The characterization of the clusters should first be done based on the variables applied for cluster analysis. Clusters can be interpreted based on cluster centroids (Means). Means were compared using analysis of variance, because the dependent variables are metric. Standard deviation values provide information on the extent to which homogeneous groups have been created.

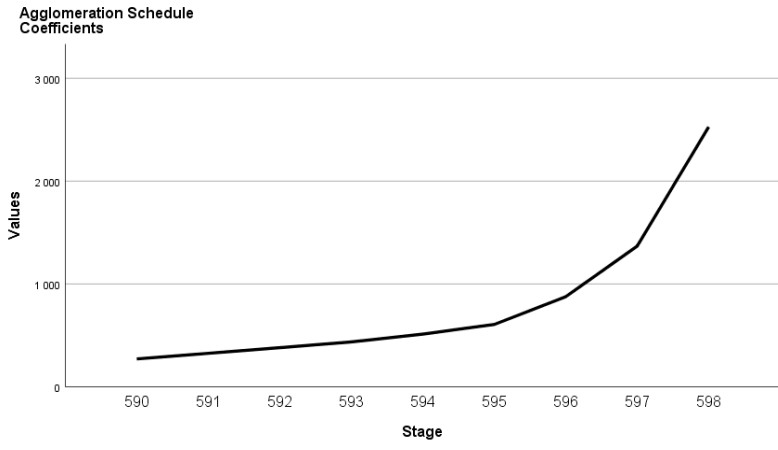

**Figure 4.** Cluster analysis—Agglomeration Schedule—Statistics: Coefficients. Source: Authors' own analysis with IBM SPSS Statistics 25.

Based on the two-cluster solution, the Std. deviation of AS_1 is quite large (1.90597), suggesting the weakness of group homogeneity (Table A2—in the Appendix A). In the case of the three-cluster solution, we can see homogeneous groups (Table 6), since the Std. deviations have decreased compared to the two-cluster solution. Based on this, we chose a three-cluster solution.

**Table 6.** Cluster analysis—Three clusters. Source: Authors' own analysis with IBM SPSS Statistics 25.

| Ward Method | | Self-Driving Bus between the Terminals (AS_1) | AI-Based Robots for Communication (AS_3) |
|---|---|---|---|
| | Mean | 4.6320 | 3.93 |
| 1 | N | 243 | 243 |
| | Std. Deviation | 0.98981 | 1.101 |
| | Mean | 6.6693 | 1.72 |
| 2 | N | 204 | 204 |
| | Std. Deviation | 0.4320 | 0.481 |
| | Mean | 1.8101 | 4.85 |
| 3 | N | 146 | 146 |
| | Std. Deviation | 0.69938 | 1.001 |
| | Mean | 5.1135 | 3.9422 |
| Total | N | 593 | 593 |
| | Std. Deviation | 1.77698 | 1.034 |

Starting from the three-cluster solution, socio-demographic variables (gender, generation) and a variable showing the evaluation of sustainability attitudes were included in the typology, which was not used in the cluster analysis but may specify the characteristics of each cluster.

According to genders, the first and the third clusters are dominated by female respondents (Figure 5). The second cluster has the highest proportion of male respondents (76.0%). To determine the correlation between variables, the $\chi2$ test was performed.

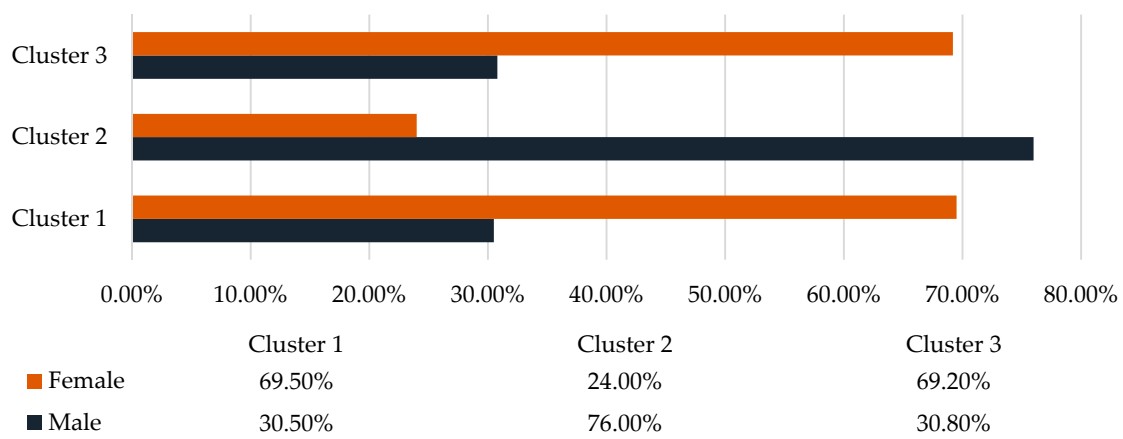

| | Cluster 1 | Cluster 2 | Cluster 3 |
|---|---|---|---|
| ■ Female | 69.50% | 24.00% | 69.20% |
| ■ Male | 30.50% | 76.00% | 30.80% |

**Figure 5.** Cross-tabulation: GENDER Ward Method. Source: Authors' own analysis.

Pearson's Chi-Square Test (Table A3—in the Appendix A) shows that the observed value of the indicator is 2.909, which, even when examined at the bilateral significance level of 0.573 (ASYMP. SIG. (2-sided)), exceeds the theoretical threshold, i.e., the significance level is lower than the selected significance level of 0.05. Consequently, we reject that there is no correlation between the two variables. Based on the Likelihood Ratio, the relationship is significant (0.460 > 0.005), so the gender distribution correlates with the clusters formed.

According to generation variable, the dominance of the Z and Y generation respondents can be observed in all clusters. Therefore, we examined how the distribution of these two age groups changes within the clusters (Figure 6). We can see that in the first cluster the ratio of the two age groups is quite balanced, and the role of the Z generation is slightly

larger. The second cluster shows the dominance of the Y generation age group, while the third cluster shows a large potency of the Z generation.

| | Cluster 1 | Cluster 2 | Cluster 3 |
|---|---|---|---|
| ■ Other generations | 20.08% | 13.70% | 17.80% |
| ■ Generation Y | 34.71% | 58.41% | 26.71% |
| ■ Generation Z | 45.21% | 27.89% | 55.49% |

**Figure 6.** Cross-tabulation GENERATION Ward Method. Source: Authors' own analysis.

By analysing the correlation between the variables, the observed value of the Pearson Chi-Square (Table A4—in the Appendix A) index is 4.456, which, even when tested at the bilateral significance level of 0.924 (ASYMP. SIG. (2-sided)), exceeds the theoretical threshold, i.e., the significance level is lower than the selected significance level of 0.05. In this case we also reject the null hypothesis that there is no correlation between the two variables. Based on the Likelihood Ratio, the relationship is significant (0.939 > 0.005), and so the age distribution is also correlated with the clusters formed.

In the first and third clusters, the importance of sustainability is below the mean value (Table 7), but in the second cluster, we can see a much higher value (Mean: 6.3695) compared with TOTAL values (Mean: 5.0101). Based on Std. Deviation, groups can be considered relatively homogeneous (values below 2.).

**Table 7.** Importance of sustainability in clusters. Source: Authors' own analysis with IBM SPSS Statistics 25.

| Ward Method | Mean | N | Std. Deviation |
|---|---|---|---|
| 1 | 4.7970 | 243 | 1.30425 |
| 2 | 6.3695 | 204 | 1.57156 |
| 3 | 4.9949 | 146 | 1.43647 |
| Total | 5.0101 | 593 | 1.45291 |

## 5. Research Outcomes

A literature review on recently published articles ($n$ = 31) indicated the high potential of AI-based airport services and the research gaps concerning passengers' perspectives. Based on the identified research gaps (e.g., the lack of surveys of the adoption of self-driving vehicles, AI robots in an airport context), we explored attitudes towards novel, less discussed applications of the technology.

Based on the analysis, we can see a higher attractiveness of transport solutions, especially those of the air-side zone (Mean: 4.70). Respondents would prefer to try self-driving vehicles for a short period in a well-regulated environment (between terminals). However, the technology might play a role in leisure activities within the air-side zone (e.g., the possibility to try self-driving cars at the airport). Although consumers have already tried AI robots at several airports, the attractiveness of the technology lags behind self-driving transportation solutions.

Based on the cluster analysis, three easily distinguished consumer groups have been created (Table 8). In the case of the first cluster, we can see that the segment is less open to technological innovation. Based on the gender and generation variables, female respondents from the generation Z are dominant who are less environmentally conscious. Based on the characteristics, the group was named "Negligents". In the second cluster, which is dominated by men, there seems to be great interest in self-driving mobility. In contrast, the cluster prefers traditional information gathering over AI robots. Men and generation Y dominate only in this cluster. Subjects are also committed to sustainable consumption. Based on the characteristics, the group was named "AV Enthusiasts". In the third cluster, the opposite attitude can be observed. Consumers (mainly female respondents from the generation Z) are less open to self-driving transport solutions but would like to use AI robots at airports to obtain information. Based on the characteristics, the group was named "Robot Fanatics".

**Table 8.** Result of cluster analysis. Source: Authors' own editing.

| Variables | | Cluster 1. | Cluster 2. | Cluster 3. |
|---|---|---|---|---|
| Variables included by cluster analysis | AS_1 Attractiveness of self-driving autobuses between terminals | below the mean | largely above the mean | largely below the mean |
| | AS_3 Attractiveness of AI-robots for information gathering | slightly above the mean | below the mean | largely above the mean |
| Variables to characterize the group | Gender | female | male | female |
| | Age | Generation Z | Generation Y | Generation Z |
| | Environmental consciousness | below the mean | largely above the mean | almost equal to the mean |
| Name of clusters | | Negligents | AV Enthusiasts | Robot Fanatics |

Based on the analysis, we answered our hypotheses and interpreted the results (Table 9).

**Table 9.** Hypotheses—results. Source: Authors' own editing.

| No. | Statement | Result | Interpretation | Method |
|---|---|---|---|---|
| H1 | *AI-based land-side transport-related services are attractive among tourists.* | **partially accepted** | *AI-based land-side services seem to be relatively attractive (Mean: 4.34), but less attractive compared to air-side services (Mean: 4.70).* | Descriptive statistics, cluster analysis |
| H2 | *AI-based air-side transport-related services are attractive among tourists.* | **accepted** | *According to descriptive statistics, AI-based air-side services are attractive among respondents (Mean: 4.70). A high attractiveness can be observed in one of the three clusters, "AV Enthusiasts".* | |
| H3 | *AI-robots are more attractive than AI-based transport services among tourists.* | **rejected** | *The attractiveness of AI-based transport services exceeds the attractiveness of AI-robots. Cluster "AV Enthusiasts" proves a significantly lower attractiveness.* | |
| H4 | *There is a correlation between "Gender" and the attractiveness of AI-based services.* | **accepted** | *Significant proportion of male respondents are open to AI-based transport solutions. Cluster "Robot Fanatics" refers to the potential of female consumer groups.* | Cluster analysis |

**Table 9.** *Cont.*

| No. | Statement | Result | Interpretation | Method |
|-----|-----------|--------|----------------|--------|
| **H5** | *There is a correlation between "Age" and the attractiveness of AI-based services.* | **accepted** | *Based on cluster analysis, as the age progresses, the attractiveness of self-driving cars decreases (Generation Z: attractiveness always below the mean), and that of AI robots increases (Generation Y: attractiveness largely above the mean).* | |
| **H6** | *There is a correlation between "Sustainable attitude" and the attractiveness of AI-based services.* | **accepted** | *Cluster analysis proved that environmentally conscious behaviour is higher among those who rate the technology positively (Clusters: AV Enthusiasts and Robot Fanatics).* | |

## 6. Conclusions

This study has been undertaken to answer the potential of AI-based services in airport development. Due to the high proportion of the respondents from the generation Z and Y, our finding for these age groups can be considered valid and useful for further research. Although the younger age group is generally considered to be the primary target group for digitized services, complementing the findings of previous research [33,35], our cluster analysis revealed differences regarding the perceived attractiveness of AI-based services among these generations.

Our results support that generation Z prefers information gathering without any human assistance, as the perceived attractiveness of AI-based technology is higher among this age group (see: Robot fanatics). At the same time, in the case of radically new technologies they do not have experience with (such as self-driving vehicles), generation Z is more cautious and less open (see: Negligents and Robot fanatics). An interesting difference in the case of gen Y is that there is more openness to self-driving vehicles among them (see: AV-Enthusiasts). Based on sociodemographic variables, our results highlighted that the primary target group for self-driving vehicles are males who belong to the millennial (gen Y) age group.

In addition to previous research on the applicability of AI-based technology [3,7], our results proved that the airport environment may be suitable for gaining experience with such services. The development of AI-based airport services can not only contribute to improving the acceptance of technology, but also to the general perception and attractiveness of airports. Enhancing the stay experience is an increasingly important goal of airports, to which experience elements based on AI technology (e.g., test driving of a self-driving car at the airport) can also contribute, starting from the positive attitude based on our results. As AI-based services are attractive to environmentally conscious consumers, linking the two directions of development can positively change the perception of the airport operation.

Overall, our outputs might be useful for practitioners, as results suggest that it might be profitable to target Z and Y generation customers during airport service development and provide them with the opportunity to try out AI-based services for fulfilling their needs for mobility, communication, and entertainment (e.g., having a conversation with an AI robot, try out a self-driving vehicle) as well.

The scientific relevance of the research comprises the discussion of AI-based solutions in a special (airport development) context (1): although an increasing number of journal articles related to airport digitization technology are appearing [8,21], the range of articles focusing on AI-based services is limited in the airport context, to the extension of which our research has contributed. With our research, the definition of airport (SST) services has been supplemented with AI-based transport and communication solutions, which is a novel approach in contrast to previous research [18,33] (2). The reinterpretation of SST can serve

as a basis for the typology of airport services as well as rethinking the factors influencing airport attractiveness. The study of the role of disruptive technologies (e.g., automation in transport, AI-based communication possibilities) in tourism is significantly limited; our results can serve as a starting point for exploring the impacts of tourism experiences on technology acceptance (3). In this context, an important direction of the research could be to model the technology acceptance of tourists, in which the impacts of the experiences gained during trips (e.g., using a self-driving bus or car at the airport—travel environment as a moderator variable in technology acceptance) can be further analysed.

## 7. Limitations

The research was not conducted on a representative sample; much of the sample consisting of the generation Z and Y age group. Therefore, further research might be needed to extend our conclusions to a wider range of population. Due to the sensitivity of consumer attitudes and the current situation of the aviation sector, it may be worthwhile to repeat the data collection after the pandemic (COVID-19), which may help to reconsider the validity of our results i.e., the potential of AI-based service development.

**Author Contributions:** Conceptualization, M.M. and M.J. and D.T.; methodology, M.M. and M.J. and D.T.; validation, M.M. and M.J. and D.T.; data curation, M.M.; writing—original draft preparation, M.M. and M.J. and D.T.; writing—review and editing, M.M. and M.J. and D.T.; visualization, M.M.; supervision, M.J. All authors have read and agreed to the published version of the manuscript.

**Funding:** This research received no external funding.

**Institutional Review Board Statement:** Not applicable.

**Informed Consent Statement:** Not applicable.

**Data Availability Statement:** The data presented in this study are available on request from the corresponding author.

**Conflicts of Interest:** The authors declare no conflict of interest.

## Appendix A

**Table A1.** Sociodemographic characteristics of subjects (%). Source: authors' own editing.

| Variables | Categories | Percentage (%) |
|---|---|---|
| Education | primary school or less | 2.13% |
| | vocational school | 1.48% |
| | secondary technical school/high school | 46.31% |
| | non-tertiary education after high school | 10.02% |
| | BA/BSc | 19.54% |
| | MA/MSc | 18.72% |
| | PhD/DLA | 1.81% |
| Social class | Lower class | 3.62% |
| | Middle class | 61.67% |
| | Upper class | 27.63% |
| | N/A | 7.17% |

**Table A2.** Cluster analysis—Two clusters. Source: Authors' own analysis with IBM SPSS Statistics 25.

| Ward Method | | Self-Driving Bus between the Terminals (AS_1) | AI-Based Robots for Communication (AS_3) |
|---|---|---|---|
| 1 | Mean | 3.9914 | 4.60 |
| | N | 345 | 345 |
| | Std. Deviation | 1.90597 | 1.086 |

**Table A2.** *Cont.*

| Ward Method | | Self-Driving Bus between the Terminals (AS_1) | AI-Based Robots for Communication (AS_3) |
|---|---|---|---|
| 2 | Mean | 6.6693 | 1.72 |
| | N | 248 | 248 |
| | Std. Deviation | 0.47140 | 0.683 |
| Total | Mean | 5.1135 | 3.9422 |
| | N | 593 | 593 |
| | Std. Deviation | 1.77698 | 1.034 |

**Table A3.** Chi-Square Tests (GENDER). Source: Authors' own analysis with IBM SPSS Statistics 25.

| | Value | df | Asymptotic Significance (2-Sided) |
|---|---|---|---|
| Pearson Chi-Square | 2.909 [a] | 4 | 0.573 |
| Likelihood Ratio | 3.617 | 4 | 0.460 |
| Linear-by-Linear Association | 0.043 | 1 | 0.836 |
| N of Valid Cases | 593 | | |

[a]. 3 cells (33.3%) have expected count less than 5. The minimum expected count is 0.26.

**Table A4.** Chi-Square Tests (GENERATION). Source: Authors' own analysis with IBM SPSS Statistics 25.

| | Value | df | Asymptotic Significance (2-Sided) |
|---|---|---|---|
| Pearson Chi-Square | 4.456 [a] | 10 | 0.924 |
| Likelihood Ratio | 4.181 | 10 | 0.939 |
| Linear-by-Linear Association | 0.140 | 1 | 0.708 |
| N of Valid Cases | 593 | | |

[a]. 4 cells (22.2%) have expected count less than 5. The minimum expected count is 0.79.

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
