# Peer review of "Technology-Enhanced Airport Services—Attractiveness from the Travelers’ Perspective"

_sustainability, doi:10.3390/su13020705_

Round 1
Reviewer 1 Report
Dear Authors,
I would recommend to Authors review this paper.
1º This need to plan the structure of paper, there are many tables and little results and conclusions. I remind the authors, that they can put the table in conclusions, please, it would go in results of research.
2º There are too many data in the paper, they have to show some figures and images to stage the results. Furthermore, I recommend some authors in Technology improved airport services, such as this author:
Florido-Benítez, L. and del Alcázar, B. (2020). Airports as ambassadors of the marketing strategies of Spanish tourist destination. Gran Tour, 21, 47–78. ➢
Florido-Benítez, L. (2020). Aeropuerto de Sevilla: un éxito de buena gestión de relación e interoperabilidad en la mejora de la conectividad aérea. Revista de Turismo Estudos e Prácticas, 5(2), 1-30.
Florido-Benítez, L. (2016). Mobile Apps: Improve Airports ́ Brand Image and Differentiate Among Competitors. ARA-Journal of Tourism Research, 6(1), 39-53.
3º. Authors have to show better conclusions, they have a total of 623 answers were received, please. These results have to show better conclusions, in order to manage better decisions in the future by airport operators.
Plagiarism: However, local labour supply is decreasing, and the number of tourists is 155 constantly increasing, it would no longer be possible to manage the airport operations without 156 AI-robots. This paragraph is mentioned in this paper Artificial Intelligence and Robotics and Their Impact on the ...https://www.ibanet.org › Document › Default
Author Response
Thank you very much for your comments and suggestions for improving our manuscript, based on which we have revised our article. We respond to the suggestions of the first-round review process, according to the structure proposed by the journal.
Point1: This need to plan the structure of paper, there are many tables and little results and conclusions. I remind the authors, that they can put the table in conclusions, please, it would go in results of research. There are too many data in the paper, they have to show some figures and images to stage the results.
Response1: Thank you very much for your recommendation. As we agree with your statement, we have made the following changes during the revision of the article:
- To prove better transparency of the manuscript, we have moved some tables to the appendix that are less related to the core results: Table 10: table of respondents’ sociodemographic traits (education, social class); Table 11: the explanation of two cluster solution; Tables (12 and 13) of Chi-Square Tests (gender and generation variable).
- To prove a better interpretation of core results, some parts of the text has been supplemented with figures: Figure 3: the explanation of respondents’ age and gender: Respondents were regrouped by age based on generational marketing, which might provide a better overview, and due to the dominance of the Z and Y generations in the sample, a more accurate interpretation was needed. The theory of generational marketing is interpreted shortly and illustrated in Figure 2.
- During the interpretation of the characteristics of each cluster group, instead of the previous tables, we interpreted the distribution of respondents based on gender and generation within the clusters with figures (Figure 5 and 6). Figures represents an important change in the interpretation of the results: during the introduction of the cluster groups, we have made findings only for the Y and Z generations due to the imbalance of the sample based on generation variable (the underrepresentation of other age groups).
Point2: Furthermore, I recommend some authors in Technology improved airport services.
Response2: Thank you very much for your recommendations. We have extended our literature review with one of the manuscripts you have suggested which proves our statement regarding the role of airport development in destination image and tourism experience.
Point3: Authors have to show better conclusions, they have a total of 623 answers were received, please. These results have to show better conclusions, in order to manage better decisions in the future by airport operators.
Response3: During the revision, we have completely rewritten the conclusion chapter to better emphasize the relevance both of our secondary and primary research findings. We pointed out that the results of our research clarify the attitude of the young age group towards disruptive technology, and in contrast to previous research, caution can also be observed among young respondents based on the results. In light of the cluster analysis, we made proposals for airports regarding AI-based technology service development (e.g., self-driving vehicles can be successful among men from the millennial age group). The academic relevance of our results has also been reinterpreted (e.g. the redefinition of SST services, interpretation of consumers’ attitude towards AI-based service in airport context, further research directions – travel/tourism environment as a moderator variable on technology acceptance of AI-based services).
Point 4: Plagiarism: However, local labour supply is decreasing, and the number of tourists is 155 constantly increasing, it would no longer be possible to manage the airport operations without 156 AI-robots. This paragraph is mentioned in this paper Artificial Intelligence and Robotics and Their Impact on the ...
Response 4: We have not used the following publication before, however, the message of the publication is in line with our statements, so we have supplemented the bibliography with this publication. Thank you very much for your suggestion.
Taking into account the suggestions, we tried to strengthen the message of our research as well as the transparency of the study. Thank you very much for your advice, we trust your positive feedback and we look forward to any further development suggestions.

Reviewer 2 Report
Title does not match the content. Different group groups were not tested - only one group selected. The title needs to be changed. There is no information about the respondents in the abstract. The information in detail that the sample is unrepresentative is not good news. I would consider giving up the questionnaire (3) of the empirical approach - I believe that this underestimates the value of the article - a non-representative sample. The argumentation of a positive attitude towards new technology is too obvious and not innovative. Footnotes are not filters and do not match the journal template (the date of using the website is missing). The article has potential, but one should consider whether the attractiveness of the "new technology" is an innovation? In my subjective opinion, it is not very innovative. Most of the respondents were women aged 18-29 - the results cannot be generalized to the entire population. The article requires thorough changes, I do not recommend publishing it in the form. An unrepresentative sample is disqualifying and lack of innovation.Author Response
Thank you very much for your comments and suggestions for improving our manuscript, based on which we have revised our article. We respond to the suggestions of the first-round review process, according to the structure proposed by the journal.
Point1: Title does not match the content. Different group groups were not tested - only one group selected. The title needs to be changed.
Response1: We have agreed with your proposal, therefore, during the revision of our manuscript, we have modified the title. We believe that the new title “Technology-enhanced airport services – attractiveness from the travelers’ perspective” is more adequate and highlights better our research focus, that is the analysis of consumers’ attitude towards novel technologies.
Point2: There is no information about the respondents in the abstract.
Response2: Thank you for drawing our attention to this shortcoming. We have supplemented the abstract with the necessary information regarding the research subjects (Line 16-17 and 23-25).
Point3: The information in detail that the sample is unrepresentative is not good news. I would consider giving up the questionnaire (3) of the empirical approach - I believe that this underestimates the value of the article - a non-representative sample. Most of the respondents were women aged 18-29 - the results cannot be generalized to the entire population.
Response3: The sample generated during the data collection is indeed not representative of the total population, which is emphasized in the Limitations chapter. However, apart from this, we decided to conduct the analysis based on the sample for the following reasons:
- The number of respondents is high (593), which exceeds the number of subjects required for exploratory marketing research in general, therefore, conclusions can be valid based on the data collection.
- However, we agree with the reviewer’s opinion that our previous findings, which applied to several age groups, are not convincing enough due to the dominance of the young age group in the sample. Therefore, based on the data collection, we have made modifications in the analysis, formulating our findings only for the dominant age groups (generation Z and Y) in the sample.
The most important further modifications regarding this topic are the following:
- During the interpretation of the characteristics of each cluster group, instead of the previous tables, we interpreted the distribution of respondents based on gender and generation within the clusters with figures (Figure 5 and Figure 6). Figures represent an important change in the interpretation of the results: during the presentation of the cluster groups, we have made findings only for the Y and Z generations due to the imbalance of the sample based on generation variable (the underrepresentation of other age groups). As a result, findings are more reliable because they only apply to the age groups that make up the sample at a high proportion (generation Z and Y).
- The findings made in the summary table of the cluster groups apply only to the two generations examined. Depending on this, the conclusion chapter has also been redesigned.
Other important changes that we kindly would like to draw the reviewer's attention to:
- To prove better transparency of the manuscript, we have moved some tables that are less related to the core results to the appendix: Table 10: table of respondents’ sociodemographic traits (education, social class); Table 11: the explanation of two cluster solution; Tables (12 and 13) of Chi-Square Tests (gender and generation variable).
- To prove a better interpretation of core results, some parts of the text has been supplemented with figures: Figure 3: the explanation of respondents’ age and gender: Respondents were regrouped by age based on generational marketing, which might provide a better overview, and due to the dominance of the Z and Y generations in the sample, a more accurate interpretation was needed. The theory of generational marketing is interpreted shortly and illustrated in Figure 2.
Point4: The argumentation of a positive attitude towards new technology is too obvious and not innovative. The article has potential, but one should consider whether the attractiveness of the "new technology" is an innovation? In my subjective opinion, it is not very innovative. The article requires thorough changes, I do not recommend publishing it in the form.
Response4: The positive attitude towards new technology may indeed seem obvious among the younger age group, but our results also showed that there are a significant number of young people who are less open to technology. During the revision, we have completely rewritten the conclusion chapter to better emphasize the relevance both of our secondary and primary research findings. We pointed out that the results of our research clarify the attitude of the young age group towards disruptive technology, and in contrast to previous research, caution can also be observed among young respondents based on the results. In light of the cluster analysis, we made proposals for airports regarding AI-based technology service development (e.g., self-driving vehicles can be successful among men in the millennials age group). The supplemented results also highlight that as the age progresses, the attractiveness of AI robot use decreases, but the attractiveness of self-driving vehicles increases. Another novelty of the research is that we examined the attractiveness of AI-based services in an airport context, which can be a basis for developing an airport development strategy (e.g., who to target and in what form some AI-based services should be applied (e.g. as a land-side or air-side service). The academic relevance of our results has also been reinterpreted (e.g. the redefinition of SST services, interpretation of consumers’ attitude towards AI-based service in airport context, further research directions – travel/tourism environment as a moderator variable on technology acceptance of AI-based services).
Point5: Footnotes are not filters and do not match the journal template (the date of using the website is missing).
Response5: Thank you for drawing our attention to this shortcoming. We have supplemented the references based on the journals’ requirements.
Taking into account the suggestions, we tried to strengthen the message of our research as well as the transparency of the study. Thank you very much for your advice, we trust your positive feedback and we look forward to any further development suggestions.

Round 2
Reviewer 1 Report
I still do not understand why the table is in the conclusions, I hope the table
ought be in the results.
Author Response
Point1: I still do not understand why the table is in the conclusions, I hope the table ought be in the results.
Response1: Thank you for your comment. During the second-round review process, we have moved Table 9 to the Results chapter.
Thank you very much for all your previous comments and suggestions for improving our manuscript.
Reviewer 2 Report
Dear Authors!
All my comments in the manuscript have been included. Congratulations on the scientific development and fine-tuning of the manuscript.
I recommend publishing the article.
Lots of health! Greetings!
Reviewer!
Author Response
Point1: Dear Authors!
All my comments in the manuscript have been included. Congratulations on the scientific development and fine-tuning of the manuscript. I recommend publishing the article. Lots of health! Greetings! Reviewer!
Response1:
Dear Reviewer,
thank you very much for all your comments and proposals for improving our manuscript. We are happy about the positive feedback. Thank you very much! We wish you good health and all the best.
Best regards,
Authors